# Xeomin^®^, a Commercial Formulation of Botulinum Neurotoxin Type A, Promotes Regeneration in a Preclinical Model of Spinal Cord Injury

**DOI:** 10.3390/toxins15040248

**Published:** 2023-03-28

**Authors:** Valentina Mastrorilli, Federica De Angelis, Valentina Vacca, Flaminia Pavone, Siro Luvisetto, Sara Marinelli

**Affiliations:** Institute of Biochemistry and Cell Biology, National Council of Research of Italy, Via Ercole Ramarini 32, 00015 Monterotondo, Italy

**Keywords:** botulinum neurotoxin, spinal cord injury, regeneration, motor recovery, sciatic static index, neuropathic pain, glial cells, mice

## Abstract

Xeomin^®^ is a commercial formulation of botulinum neurotoxin type A (BoNT/A) clinically authorized for treating neurological disorders, such as blepharospasm, cervical dystonia, limb spasticity, and sialorrhea. We have previously demonstrated that spinal injection of laboratory purified 150 kDa BoNT/A in paraplegic mice, after undergoing traumatic spinal cord injury (SCI), was able to reduce excitotoxic phenomena, glial scar, inflammation, and the development of neuropathic pain and facilitate regeneration and motor recovery. In the present study, as proof of concept in view of a possible clinical application, we studied the efficacy of Xeomin^®^ in the same preclinical SCI model in which we highlighted the positive effects of lab-purified BoNT/A. Data comparison shows that Xeomin^®^ induces similar pharmacological and therapeutic effects, albeit with less efficacy, to lab-purified BoNT/A. This difference, which can be improved by adjusting the dose, can be attributable to the different formulation and pharmacodynamics. Although the mechanism by which Xeomin^®^ and laboratory purified BoNT/A induce functional improvement in paraplegic mice is still far from being understood, these results open a possible new scenario in treatment of SCI and are a stimulus for further research.

## 1. Introduction

Botulinum neurotoxins (BoNTs) are produced by *Clostridium botulinum* in eight different serotypes, named by letters from A to G, and X [1,2]. A number of different subtypes, together with chimeric molecules, complete the large family of these toxins [3,4]. All serotypes consist of a 100 kDa di-chain molecule, called heavy chain (HC), which binds to nerve membrane receptors, and a 50 kDa molecule, called light chain (LC), which enters the cytosol where it cleaves the soluble NSF (N-ethylmaleimide-sensitive factor) attachment receptor (SNARE) proteins, the key components whose integrity is required for the formation of the protein complex responsible for the fusion of synaptic vesicles with the cell membrane [1,5,6,7]. All BoNTs act by specifically cleaving a different peptide bond on one of the SNARE proteins: BoNT/A and /E cleave SNAP-25; BoNT/B, /D, /G and /F cleave VAMP/synaptobrevin; whereas BoNT/C cleaves both syntaxin and SNAP-25 [8].

Several studies evidenced the use of BoNTs, mainly BoNT/A and /B serotypes, for therapy in a variety of human diseases [9,10]. Today, BoNT/A and /B are licensed for treatment of several autonomic nervous system and movement disorders, such as dystonias, muscle spasms, spasticity, excessive sweating, overactive urinary bladder, along with many off-label uses in other neurological pathologies [11,12], besides the well-known treatment for aesthetic purposes [13]. A potential role of BoNT/A as a novel agent in pain relief has also been demonstrated [14,15,16,17,18,19,20], and the use of BoNT/A for the prophylactic treatment of migraine has recently been approved [21,22,23].

In a previous study [24], we demonstrated the beneficial effect of spinal injection of lab-purified 150 kDa BoNT/A protein in a mouse model of spinal cord injury (SCI) at two different degrees of severity [25,26]. In severe SCI, traumatic spinal cord injury resulted in complete hindlimb paralysis, while in moderate SCI, the damage was partial and the mice retained some movement and sensation in the hindlimbs. Within one hour from SCI, i.e., during the acute phase of the injury, we injected a single dose of BoNT/A (15 pg dissolved in 5 μL of saline) and observed: (i) an extraordinary motor recovery from paralysis with reconstruction of the damaged spinal cord in mice with severe SCI; and, (ii) in addition to motor recovery, a prevention of the development of neuropathic pain, a comorbidity often associated with SCI, in mice with moderate SCI.

The purpose of the present study was to investigate whether the beneficial effects of lab-purified BoNT/A in counteracting SCI [24] could also be obtained using a commercial formulation of BoNT/A. Among the various formulations of BoNT/A on the market, we chose Xeomin^®^ (IncobotulinumtoxinA; Merz Pharmaceuticals GmbH, Frankfurt am Main, Germany) because this drug does not contain complexing proteins, which keeps the molecule similar to the one purified in the lab.

## 2. Results

To evaluate the effects of Xeomin^®^ on motor paralysis, we tested mice with both severe and moderate SCI; to evaluate the effect of Xeomin^®^ on sensory deficits and neuropathic pain symptoms, we tested only mice with moderate SCI. Furthermore, to avoid the entry of toxin into the blood stream, as a consequence of the destruction of the blood-spinal barrier by SCI at the thoracic level (T9–T11), we injected Xeomin^®^ (single dose of 2.5 U dissolved in 5 μL of saline) not directly to the lesion site but at the lumbar level (L4–L5). The dose of toxin was chosen on the basis of the conversion ratio suggested by the manufacturer: 2.5U of Xeomin^®^ corresponds to approximately 15 pg of lab-purified 150 kDa BoNT/A, a single dose used in our previous study [24]. As the control group, some SCI mice were injected with saline (0.9% NaCl).

Figure 1A shows the BMS score, expressed as incremental recovery (ΔBMS) starting from day 3 after SCI (D3), obtained in severe SCI mice treated with saline or Xeomin^®^. As inclusion criteria, all mice that at D3 performed a BMS score in the range 0–3 were considered in this group. At D3, ΔBMS was calculated with respect to baseline value of BMS = 9 (normal movement of the hindlimbs) and the days from SCI, which in our experimental protocol is equal to 3, by using the Equation (1): ΔBMS at D3 = ((BMS value at D3 − 9)/3)(1)
while at Dx (x = 7, 10, 14, 21, 28, 35), ΔBMS was calculated with respect to BMS value at D3 and the days from D3, by using the Equation (2):ΔBMS at Dx = ((BMS value at Dx − BMS value at D3)/days from D3)(2)

Although an increasing trend was observed, the mean ΔBMS of saline- or Xeomin^®^-treated SCI mice was not different from D3 to D21. Starting from D28, Xeomin^®^-treated mice showed a significant amelioration of motor deficit when compared with saline-treated mice (ANOVA for repeated measures: interaction time x treatment F_6,54_ = 7.557, *p* < 0.0001) (see also Appendix A). 

Figure 1B shows the ΔBMS score, as a function of days after SCI, obtained in moderate SCI mice treated with saline or Xeomin^®^. All mice that at D3 performed a BMS score in the range of 3 to 6 were included in this group. Both saline- and Xeomin^®^-treated mice were not significantly different.

Because the hindlimbs of moderate SCI mice were not completely paralyzed, these mice retained nociceptive sensitivity and thus could respond to nociceptive stimuli. For this reason, on moderate SCI mice, we were able to perform a behavioral analysis to ascertain whether Xeomin^®^ was able to counteract the onset of neuropathic pain. Figure 1C,D shows the percentages of mechanical allodynia threshold (panel C) and thermal hyperalgesia threshold (panel D) in moderate SCI mice, at different days post SCI, calculated for each mouse with respect to the baseline threshold before SCI (BL) using Equation (3):% of threshold = (threshold at Dx/BL) × 100(3)

Both saline- and Xeomin^®^-treated mice developed and maintained mechanical allodynia for all time-points considered. ANOVA one-way comparing treatments with their baseline shows a significant effect for both the saline (F_5,50_ = 13.645, *p* < 0.0001) and Xeomin^®^ (F_5,46_ = 8.685, *p* < 0.0001) group. Post-hoc comparison between groups evidenced that Xeomin^®^ was significantly different from saline at D28.

More evident is the effect of Xeomin^®^ on thermal hyperalgesia. In detail, the threshold-to-thermal stimuli in saline-treated SCI mice was significantly reduced, indicating greater sensitivity to thermal pain, compared with baseline values at each testing day. In contrast, the thermal pain response in Xeomin^®^-treated SCI mice was not significantly different from baseline values. One-way ANOVA comparing treatments with their baseline shows a significant effect for saline (F_5,49_ = 13.336, *p* < 0.0001) but not for the Xeomin^®^ group. This finding indicates that Xeomin^®^ was able to prevent the worsening of thermal sensitivity due to the development of neuropathy induced by SCI.

Figure 1E reports the Sciatic Static Index (SSI) calculated from the analysis of footprint parameters in saline- or Xeomin^®^-treated moderate SCI mice. Figure 1F shows two representative examples of footprint during walking. The SSI allows to evaluate motor function and ambulation recovery because it analyzes footprint parameters not easily detectable with BMS analysis. The SSI analysis shows that saline- and Xeomin^®^-treated moderate SCI mice had an approximately 35% walking deficit at D3 after SCI, and only Xeomin^®^ treatment was able to induce rapid and significant recovery. Two-way ANOVA for repeated measures shows a significant main effect for treatment (F_1,18_ = 7.18, *p* = 0.015) and time (F_3,5_ = 3.69, *p* = 0.0044). Moreover, note that the footprint of Xeomin^®^-treated mice at D28 and D35 showed a regular gait compared with saline-treated mice.

Figure 2 shows examples of immunofluorescence (IF) images of glial fibrillary acid protein (GFAP, astrocyte marker) expression in spinal cord sections taken from the impact zone in saline- or Xeomin^®^-treated severe SCI mice, at D35 after SCI. As previously demonstrated [24], large areas of the spinal cord of saline-treated SCI mice were damaged with glial scarring and astrocyte hyperactivation, and the spinal cord size was markedly reduced with the spinal horns completely enveloped by the glial scar. In the Xeomin^®^ treated SCI mouse, although astrocyte activation and small glial scarring areas were still present, especially in the impact zone, the spinal cord morphology was preserved, with clearly recognizable dorsal and ventral horns.

To test whether Xeomin^®^ could exert a neuroprotective action against excitotoxic phenomena, as previously observed for lab-purified BoNT/A [24], and considering that glutamate induces excitotoxic cell death [27], we analyzed the expression of vesicular glutamate transporter type 1 (vGLUT1) in colocalization with GFAP expression in severe SCI mice, at D35 after SCI. Figure 3A,B shows representative examples of IF images of the colocalization of vGLUT1 with GFAP expression in spinal cord sections taken from the impact area. In the spinal cord of saline-treated SCI mice (panels A), a striking expression of vGLUT1 was detected together with a strong colocalization with GFAP. In contrast, in spinal cord of Xeomin^®^-treated mice (panels B), both vGLUT1 expression and its colocalization with GFAP were strongly reduced.

## 3. Discussion and Conclusions

In a previous article [24], we demonstrated that a spinal injection of lab-purified 150 kDa BoNT/A was able to: (i) induce spinal nerve regeneration and motor recovery in a severe SCI mice model; and (ii) counteract the development of neuropathic pain in a moderate SCI mice model. These effects correlated with reduced activation of spinal glia and reduced formation of glial scar, events that positively contrast the cascade of adverse phenomena occurring during SCI-induced secondary injury [28,29,30,31,32]. Thus, by modulating the glial response, BoNT/A allows a gradual progression toward a facilitated recovery of motor and sensory function.

In the present study, we tested Xeomin^®^ to evaluate its possible therapeutic potential in counteracting SCI with a view to its possible clinical application. We chose Xeomin^®^ because, of all the BoNTs available on the market, it is the commercial formulation that most closely matches the lab-purified BoNT/A. Although our findings unequivocally demonstrate a therapeutic effect of Xeomin on SCI, we also highlighted some substantial differences in comparison with the effects of lab-purified BoNT/A. The most evident regards the pharmacokinetic. In fact, in comparison with the lab-purified BoNT/A, we observed a delay in the regenerative and neuroprotective effects on SCI with Xeomin^®^, as well as modest effectiveness. In detail, the time of onset of therapeutic effects was the first day after administration in the case of lab-purified BoNT/A [24], while in the case of Xeomin^®^, it was three weeks after administration.

Another discrepancy concerns motor recovery, estimated by the BMS parameter. In fact, motor recovery was almost complete in the case of lab-purified BoNT/A, with BMS values between 8 and 9 [24], while motor recovery was not complete in the case of Xeomin^®^, with BMS values between 6 and 8. This is particularly evident in the moderate SCI model in which it was not possible to highlight better motor recovery of Xeomin^®^ compared to saline. This is probably caused by the ability of the murine axons to regenerate. When the neurological insult is moderate and does not completely affect the spinal nervous system, the natural functional recovery can mask the limited therapeutic effect of Xeomin^®^. It is also evident that, although Xeomin^®^ appears to be ineffective in moderate SCI, analysis of footprints during walking demonstrates a strong ability of Xeomin^®^ to induce early functional recovery.

Comparing the results obtained from lab-purified BoNT/A [24] with results obtained from Xeomin^®^, another difference immediately emerges in the effects on neuropathic pain. In fact, although Xeomin^®^ was able to mitigate the symptoms of neuropathic pain, it did not completely prevent the onset of neuropathic pain or restore the physiological threshold, especially in mechanical allodynia. On the other hand, from an overview of spinal morphology, astrocytosis and excitotoxicity, Xeomin^®^ was effective in inducing neuroprotection and reduction of glial scar.

All these discrepancies can be reasonably attributed again to pharmacodynamics. It has been already demonstrated that different BoNT/A formulations influence the effectiveness requiring a dose adjustment [33,34,35]. In particular, Byun et al. [35] evidenced that BoNT/A with high efficacy and duration is that one lacking human-derived component (coretox^®^), such as human albumin, which are substituted by polysorbate 20 and L-methionine. Also the higher efficacy of our lab-purified BoNT/A, with respect to Xeomin^®^, could be explained by the absence of these animal components. This is an important point to be considered because pharmacodynamics and pharmacocinetics of the toxin are influenced by the immunesystem response that can be activated by the exogenous antigens, such as the effect exerted by human albumin in mice.

Moreover, the different onset of therapeutic effects of Xeomin^®^, the BoNT/A formulation used in this study, compared to lab-purified 150 kDa BoNT/A protein used in [24] may originate from the difficulty obtaining the exact correspondence between doses of toxin, expressed by units of Xeomin^®^ and pico-grams of BoNT/A in [24]. Dose correspondence between picograms and units of Xeomin^®^ was calculated considering that vials of Xeomin^®^ contained approximately 600 pg of toxin per 100 units, thus 15 pg of lab-purified 10 kDa BoNT/A corresponded to 2.5 units of Xeomin^®^. Another discrepancy between the two BoNT/A formulations resides in the solvent and/or excipients used in preparation of lab-purified BoNT/A and Xeomin^®^. Both BoNT/A are prepared by dilution of stock solution in saline (0.9%), but Xeomin^®^ also contains human albumin and sucrose.

Overall, our results demonstrate that Xeomin^®^ may be a possible candidate in clinical application as a therapy against SCI, although a dose-response study would be desirable. In conclusion, the present study represents proof of concept for the clinical application of BoNT/A in the therapy of traumatic SCI, as we validated and confirmed the pro-regenerative and neuroprotective action of Xeomin^®^, one of the most used commercial compounds in neurological disorders.

## 4. Materials and Methods

### 4.1. Animals

Four-month-old CD1 female mice (EMMA Infrafrontier, Monterotondo, Italy) were used. Mice were housed in groups of 4 in standard cages under a 12/12 h light/dark cycle (7:00 a.m.–7:00 p.m.), with food and water available ad libitum. Thirty minutes before surgery, mice were moved to a surgical room and were randomly assigned to different experimental groups. The groups’ size for in vivo experiments was calculated by implementing a power analysis (Gpower 3.1), and the number of mice used is reported in the figure legends. Testing was done by blind investigators as for treatment groups. Care and handling of mice were in accordance with the guidelines of the Committee for Research and Ethical Issues of IASP [36]. In vivo procedures were approved by the Italian Ministry of Health (PR122/2019, 10 February 2019) on the use of animals for research.

### 4.2. Surgery

Different groups of mice were subjected to SCI with severe or moderate traumatic injury. Detailed surgical procedures and postoperative care were as described in [24,25]. Spinal cord contusion was done at the thoracic level (T9-T11) using a cortical PinPoint precision impactor device (Hatteras Instruments Inc., Cary, NC, USA). Impactor parameters were as follows: middle, round, and flat tip (#4); depth: 5 mm; velocity: 3 m/s (severe SCI) or 1 m/s (moderate SCI); dwell time: 800 ms (severe SCI) or 75 ms (moderate SCI).

### 4.3. Drugs

Within 1 h from contusion, a single dose (2.5U in 5 μL saline) of Xeomin^®^ (Merz Pharmaceuticals GmbH, Frankfurt am Main, Germany), or saline (5 μL), was spinally injected through intervertebral space at the lumbar level (L4–L5), using an automatic injector (KDS 310 Plus; KD Scientific; Holliston, MA, USA) equipped with a 10 μL syringe (Hamilton #701; Biosigma; Cona, Italy) with a needle of 30 μm internal diameter (Eppendorf, Hamburg, Germany). Injections were made at 2 μL/min and, to avoid liquid reflux, the needle was maintained in situ for 3 min after the end of the injection, as already described in [24,25].

### 4.4. Behavioral Test: Basso Mouse Scale (BMS)

For all mice groups, the hindlimb functions were assessed by estimating the BMS score in an open field arena as previously described [24]. The BMS score is a parameter, ranging from 0 to 9, where 0 indicates complete paralysis while 9 indicates normal hindlimbs movement. Each mouse was tested at D3, D7, D10, D14, D21, D28, and D35 post-SCI, and BMS scores of left and right hindlimbs were averaged together. Mice with a BMS score between 0 and 3 at D3 were included in the severe SCI group, while mice with a BMS score between >3 and 6 at D3 were included in the moderate SCI group. Mild-lesioned mice with BMS > 6 were excluded. To better appreciate amelioration in BMS values, we calculated the incremental BMS performance for the first day of BMS measurement (D3).

### 4.5. Nociceptive Tests

Mechanical allodynia was tested using a Dynamic Plantar Aesthesiometer (Model 37,400, Ugo Basile Srl, Comerio, Italy) as described in [37]. Thermal hyperalgesia was tested using an automatic plantar test instrument (Plantar Test, Ugo Basile, Comerio, Italy) as described in [38]. For the thermal hyperalgesia test, a cut-off time of 15 s was imposed to avoid damage of hindpaw skin tissue. Both the mechanical and thermal threshold were measured before injury (baseline value) and at D3, D7, D10, D14, and D28 after SCI, on the same moderate SCI mice, with an interval between the two tests of one hour. For each mouse, two values of mechanical and thermal threshold after SCI were obtained because the two hindpaws, the right and the left, can develop different degrees of neuropathic pain. At each testing day, threshold values were averaged from 3 consecutive measurements per hindpaw and reported as a percentage with corresponding the baseline value.

### 4.6. Sciatic Static Index

As well as the BMS score, recovery of the hindpaw functionality was also analysed by the the recording of the walking footprints in moderate SCI mice, as described in [39]. Footprints were collected by painting with ink on the plantar surface of the hindpaws. The toe spread (TS), from the 1st to 5th toe, and the paw length (PL), from the tip of the 3rd toe to the most posterior aspect of the paw, were considered to calculate the Sciatic Static Index (SSI). These parameters were measured from at least five footprints, taken from three different walking tracks. At each different walking track, the plantar surface of the hindpaws were repainted with ink, to avoid that footprint becoming weaker after a few runs. As suggested by [40], SSI was calculated by the equation:SSI = +101.3 × (ITS − NTS)/NTS − 54.03 × (IPL − NPL)/NPL − 9.5  where ITS and NTS are the injured and normal toe spread, respectively, while IPL and NPL are the injured and normal paw length, respectively. SSI values range between 0 (normal fuction) to 100 (complete loss of hindpaw functionality).

### 4.7. Immunohistochemistry and Confocal Images

Spinal cords from saline- or Xeomin-treated animals were harvested at D35 for IF analysis as described in [24]. Briefly, sacrify mice were immediately perfused with saline and 4% paraformaldehyde (PFA) in phosphate buffer solution (PBS; pH 7.4). The spinal cord was removed, maintained 48 h in 4% PFA in PBS and, after cryoprotection with a solution of 30% (*w*/*v*) sucrose in PBS, conserved at −80 °C. A selection of T9–T11 tissue slices (40 microns) were collected in PBS up to the IF procedure. For double IF staining, slices were incubated 48h with the following primary antibodies diluted in PBS with 0.3% Triton (Sigma-Aldrich St. Louis, Missouri, USA): (i) anti-GFAP (astrocytes marker; mouse monoclonal 1:100; Sigma-Aldrich); (ii) anti-vGLUT1 (vesicular glutamate transporter 1; guinea pig 1:200; Millipore AB5905). After three washes in PBS, the slices were incubated 2 h at room temperature with ALEXA Fluor 488 donkey anti-mouse (1:100; Jackson ImmunoResearch Europe Ltd, Cambridge, UK) and Rhodamine goat anti-guinea pig (1:100; Jackson ImmunoResearch Europe Ltd, Cambridge, UK). After 2 washings in PBS, slices were incubated 10 min with bisBenzimide, DNA-fluorochrome (Hoechst, 1:1000, Sigma-Aldrich Italia, Milano, Italy) in PBS.

Low (10×) and high (40×) magnification IF images were recorded by a confocal microscopy using a TCS SP5 microscope (Leica Microsystems Srl, Milan, Italy), in sequential laser scanning mode, to rule out cross-bleeding between channels. Images were analysed by I.A.S. software (Leica Microsystems Srl, Milan, Italy).

### 4.8. Data Analysis

Experimental data are expressed as mean ± sem. Group comparisons were conducted by one-way or two-way ANOVA for repeated measures or by Student’s *t*-test. Post hoc comparisons were made with the Tukey–Kramer (statistical significance at *p* < 0.05). Data analysis was performed by StatView SAS (version 5.0, Cary, NC, USA).

## Figures and Tables

**Figure 1 toxins-15-00248-f001:**
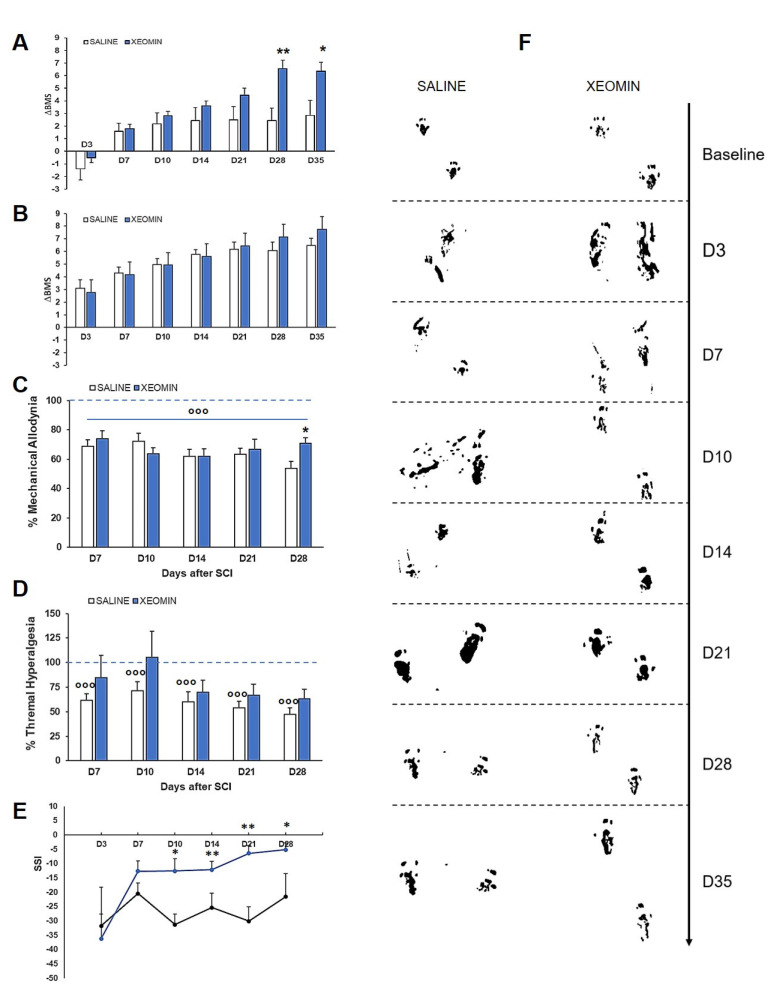
(**A**) Incremental ratio (ΔBMS) of motor recovery, calculated with respect to baseline for D3 (first day of the BMS measurement) or D3 for the subsequent days, in saline- or Xeomin^®^-treated severe SCI mice (N = 5–6/group). Statistics: Tukey-Kramer post-hoc test: (*) *p* < 0.05, (**) *p* < 0.001 vs. saline. (**B**) ΔBMS of motor recovery in saline- or Xeomin^®^-treated moderate SCI mice (N = 5/group). (**C**) Percentage of mechanical allodynia threshold in saline- or Xeomin^®^-treated moderate SCI mice (N = 5/group) with respect to the threshold before SCI. Statistics: Tukey-Kramer post-hoc test: (°°°) *p* < 0.0001 vs. baseline (blue dashed line at 100%); unpaired t-test at D28: (*) *p* < 0.05 vs. saline; (**D**) Percentage of thermal hyperalgesia threshold in saline- or Xeomin^®^-treated moderate SCI mice (N = 5/group) with respect to the threshold before SCI. Statistics: unpaired *t*-test: (°°°) *p* < 0.0001 vs. baseline (blue dashed line at 100%). (**E**) Sciatic Static Index (SSI) calculated from hindpaws’ footprints in saline- (black line) or Xeomin^®^-treated (blue line) moderate SCI mice (N = 5/group). Values of SSI = 0 represent normal walking, while negative values are an index of walking deficits. Statistics: Tukey-Kramer post-hoc test: (*) *p* < 0.05, (**) *p* < 0.001 vs. saline. (**F**) Representative examples of footprint walking track in saline- or Xeomin^®^-treated moderate SCI mice.

**Figure 2 toxins-15-00248-f002:**
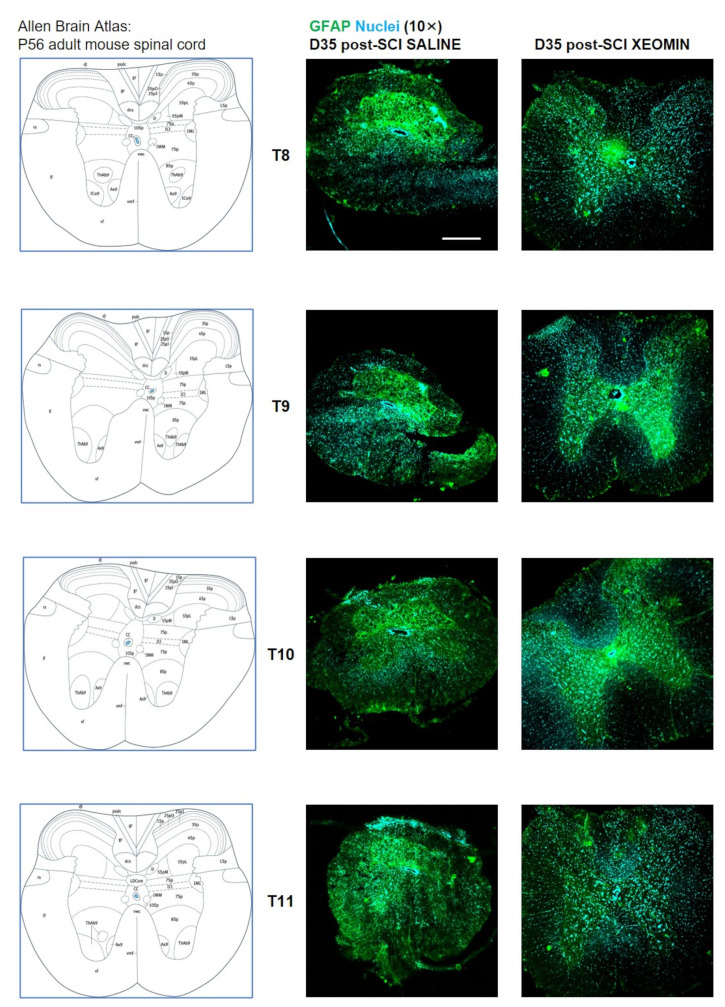
Neuroprotective effects of Xeomin^®^ on injured spinal cord. Representative examples of confocal images (10×) deriving from thoracic (T8/T11) spinal cord of saline- or Xeomin^®^-treated mice, collected at D35 after severe SCI. GFAP (green) marker evidences astroglia reactivity and/or glial scar. With respect to the impact: EPI indicates the epicenter zone of injury, i.e., the core lesion at T9, while PERI indicates the peri-lesioned zone of injury. Bar 300 μm.

**Figure 3 toxins-15-00248-f003:**
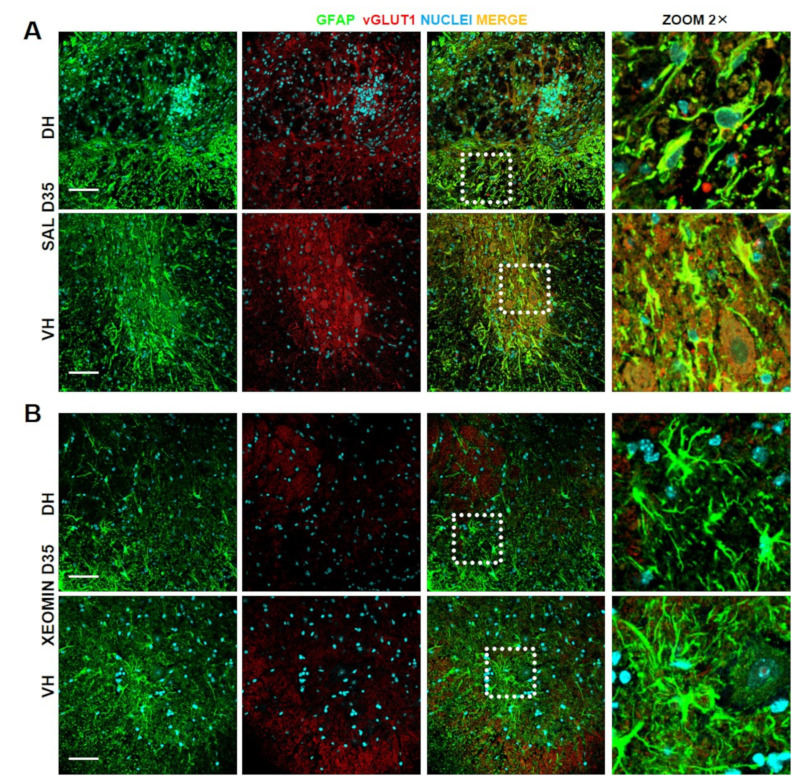
Xeomin^®^ modulates astrocytes reactivity and glial scar formation. Representative examples of confocal images (40× and 40×-zoom2×) deriving from thoracic (T9) spinal cord of: (**A**) saline- or (**B**) Xeomin^®^-treated mice, collected at D35 after severe SCI. Immunostaining in dorsal (DH) and ventral horn (VH): GFAP (green) marker stains for astrocytes gliosis and glial scar, vGLUT1 (red) marker stains for the glutamate transporter as indirect index of glutamate release, while merge (yellow) indicates colocalization of the two markers. Dotted white boxes indicate areas where the zoom was taken. Bar 60 μm.

## Data Availability

Data presented in this study are available on request from the corresponding authors.

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
