# Peer review of "Xeomin®, a Commercial Formulation of Botulinum Neurotoxin Type A, Promotes Regeneration in a Preclinical Model of Spinal Cord Injury"

_toxins, 2023, doi:10.3390/toxins15040248_

Round 1

Reviewer 1 Report

The manuscript appears to be a sound piece of research showing the effects of Xeomin® on spinal injury modelled in mice. The authors detected a difference in motor recovery compared to laboratory-purified BoNT/A. The propose two possible explanations - either based on differences in the amount of the toxin between two preparations OR on the composition of the commercial Xeomin®. It would be important to address this in more detail in Discussion, with references, listing the expected picogram amounts per drug Units and the protein composition which apparently includes albumin.

There are several minor errors to be corrected:

Page 1. l.36    Each BoNTs act   - better All BoNTs act or Each BoNT acts

Page 2, l.53 as this drug does not contain accessory proteins, such as binding albumin –

- what do you mean by ‘binding albumin’, why to use ‘proteins’ in multiples?

Page 2, l.84 significant amelioration in motor recovery – do you mean ‘motor deficit’?

Page 7, l. ..71 At difference – better ‘In contrast ,’

Page 7, l…74 that an ith injection of – needs correction

Page 7, l. ..78 spinal glial – should be ‘spinal glia’?

Author Response

Reply to Reviewer #1

We thank the reviewer for positive consideration of our paper and for helpful comments to improve its quality. All the raised concerns have been considered and appropriately replied as follow:

- General comments

".....It would be important to address this in more detail in Discussion, with references, listing the expected picogram amounts per drug Units and the protein composition which apparently includes albumin."

RE: we agree with the reviewer and these arguments have been amply discussed at the end of Discussion, from line 217 at the bottom of page 7 to line 236 of page 8.

- "There are several minor errors to be corrected"

RE: We thank the reviewer for its detailed analysis of the text; all minor errors have been corrected according to the suggestions.

Reviewer 2 Report

Manuscript “Xeomin®, a commercial formulation of botulinum neurotoxin type A, promotes regeneration in a preclinical model of spinal cord injury” reported that in a mouse contusion spinal cord injury model Xeomin could reduce excitotoxic phenomena, glial scar, inflammation, and the development of neuropathic pain and to facilitate regeneration and motor recovery. Yet, in comparison with the lab-purified botulinum neurotoxin type A from a study performed in the same laboratory the efficacy of Xeomin was less. In general, the experiments were carefully designed and performed, and the results were reliable based on the data presented in the manuscript. However, there are some concerns that need to be considered or addressed in relation to the experimental methods, results presentation, and the discussion.

1.      As pointed out by the authors, the Xeomin dose applied might be not optimal, and it requires an adjustment to achieve the best results. The authors only chose one dose: 2.5U dissolved in 5 µl saline. I wonder on what basis the authors decided to choose this dose but not tested several different doses. Also, the route to apply the drug raised another concern to me. The authors just described to inject the solution into the lumbar spinal cord. How this being done was not described in detail. Was the drug injected through a laminectomy or through the intervertebral space? Where exactly the injection was located? Was it in the gray matter or the white matter? How deep was the injection? Injection of 5 µl liquid in 2.5 min seemed too fast considering the small size of the mouse spinal cord. In that case, most of the liquid might leak out of the parenchyma of the spinal cord and to the surrounding subarachnoid space, which resulted in an injection equal to intrathecal injection.

2.      The walking pattern was not clearly described both in the Methods and in Figure 1F. How were footprints collected? Were they collected using an automatic devise or just by painting the paws with ink? If they were ink-painted, after a few runs some footprints would become weaker than the others. The footprints from Figure 1F seemed not all from the hindpaws because the shape and the sizes were not similar.

3.      The effect of Xeomin was too good to believe if you compare the morphology of the spinal cord treated with saline and Xeomin in Figure 2. I fully understand that the authors just wanted to present “good” representatives in each group. However, imagining that if the shape of the spinal cord in a Xeomin-treated group was like the shape in a saline-treated group in the beginning (just after the injury) could it become the shape presented in Xeomin-treated group now in the figure just after 35 days’ treatment? To me, what the pictures presented in the figure in saline-treated and Xeomin-treated groups should not belong to the same severeness of the injury in the beginning.

4.      It is hardly to know how many animals were use. It was said that the numbers were indicated in the figures, but it is difficult to find out. For example, in Figure 1, were the same animals used in the different subgroups/treatments (different days after the injury) or they were different animals? Were the same animals used in different measurements (e.g., BMS and mechanical allodynia)?

5.      The Discussion needs to be expanded to cover, e.g., what caused the different efficacy of lab-purified BoNT/A and Xeomin in addition to pharmacodynamic and dose. Why did you study the effect of Xeomin? Was this because Xeomin is a commercially available drug and easier to be approved in clinical application to treat spinal cord injured patients than lab-purified BoNT/A?

6.      Some minor issues:

a.      In the beginning of the Results (line 65) the authors said: “To evaluate the effects of Xeomin® on motor paralysis we tested mice with severe SCI”, but it was also test in moderate SCI as shown in Figure 1B.

b.      In Figure 1C, it is a bit difficult to understand “ooo” above a line. Please explain it a bit clearer.

c.       In Figure 1E, please indicate the blue line is for Xeomin-treated animals and black line is for saline-treated animals.

d.      Figure 2, the figure showed spinal cord sections from 4 different segment (T8-T11). Please indicate where the epicenter of the injury is and where the pericenter is.

e.      Figure 3, please make the dotted-line boxes clearer. The dots look like cell nuclei if don’t read the figure legend.

f.        Line 216, “difficult” should be “difficulty”.

g.       Line 304, “deriving” should be “derived”.

h.      Line 312/313, “…diluted in Triton 0.3%” should be “…diluted in PBS with 0.3% Triton”

Author Response

Reply to Reviewer #2

We thank the reviewer for positive consideration of our paper and for helpful comments to improve its quality. All the raised concerns have been considered and appropriately replied as follow:

- point 1 "....I wonder on what basis the authors decided to choose this but not tested several different doses..."

RE: We decided to use the dose of 2.5U of Xeomin dissolved in 5 microlt of saline because this is the dose corresponding to the 15 pgtox of lab-purified BoNT/A that we previously demonstrated to be efficacious in treatment of SCI in mice. This concentration (i.e. 15 pgtox) was chosen based on past experience and publications of our lab (see Luvisetto et al, 2006, Brain Research 1082, 124-131; Marinelli et al, 2010, Neuroscience 171, 316-328; Marinelli et al, 2012, PlosOne 7, e47977; Vacca et al, 2012, Brain Behav Immun 26, 489-499; Luvisetto et al, 2015, Toxicon 94, 23-28) where we took in consideration the comparison between doses of our BoNT/A with the Units of Botox that are considered effective doses for clinical application. A sentence has been added in Results (lines 71-73) and Discussion (lines 230-236).

- point 1 "...Also the route to apply the drug raised another concern to me..."

RE: The route of administration, intraspinal, has been chosen for different reasons. First, we have already demonstrated (Vacca et al, 2020, Toxins 12, 491) that BoNT/A directly injected at lumbar level in the spinal cord is able to have a direct and long-lasting beneficial effect on spinal neuroprotection. In fact, it is retrogradely transported to the injury site, acting along the entire zone involved in the spinal neurodegeneration and neuroinflammation, playing a fundamental role in neuroprotection of both damaged and, above all, non damaged areas thus preventing further neuronal death. Second, intraspinal injection is easily translated into clinical application since it is a routinary procedure utilized in anesthesia. Finally, we chose this way of administration also to avoid systemic effects of the toxin. In fact, we injected toxin in the lumbar zone because in the traumatic area there are ischemic events and the brain blood barrier is disrupted.

Regarding the way of the administration, we have better described it in Method. Anyway, as we have already reported in the manuscript, the SCI model utilized is without laminectomy, the cannula-over-needle device is inserted intervertebral at lumbar level. The micro-needle (for microinfusion pump) utilized is 3 mm long because we considered about 1,5 mm to pass the muscle and vertebrae thickness and about 1.5 mm to pass subarachnoid space and to arrive in the middle of the spinal cord. The dimension of the spinal cord of mice was previously measured with a microCT, as reported here and in a previous paper (Vacca et al, 2020, Toxins, 12, 491).

On the basis of this measurement and the positioning of the needle, we can assume that the injection should be in the gray matter.

Finally, regarding the question on the volume/time injection we used the lower ratio available by the micro-infusion pump, according to the needle diameter. In any case, the injection of a volume of 5 microlt in a time of 2.5 min means 2 microlt gradually and slowly injected in 1 min, thus it cannot be considered a shocking injection and surely not a bolus injection. From our experience in icv and ith injections (Luvisetto et al, 2006, Brain Research 1082, 214-131; Marinelli et al, 2010, Neuroscience 171, 316-328) this appeared to us to be an appropriate volume/time injection ratio.

- point 2 "The walking pattern was not clearly described in the Methods and in Figure 1F..."

RE: The walking pattern was better described in the Methods of the revised manuscript. The footprints were collected by painting the hindpaws with ink. The footprint parameters used to calculate SSI were taken from five different footprints from three different walking tracks. Among different walking tracks the plantar surface of hindpaws were repainted with ink, precisely to prevent the footprint from becoming weaker. Difference in the footprint shape is a consequence of injury. In saline SCI mice, as the reported example in Figure 1F, some footprints are hardly recognizable because the mouse badly rests its paw and drags it on the floor. This can also be observed from the attached videos as supplementary information.

- point 3 "The effect of Xeomin was too good to believe if you compare the morphology of the spinal cord treated with saline and Xeomin in Figure 2"

RE: As observed by the reviewer, there is a strong discrepancy between the morphology of spinal cord of mice treated with saline and Xeomin: this is strongly evident. To explain this, it must be considered that in the case of saline, the neurodegeneration progressed into a chronic phase (remember that tissues collection is 35 days after the SCI) with all the severe components that featured this period, such as cystic cavity, glial scar and further apoptosis extended to non injured areas. In contrast, in the case of Xeomin the effective and long-lasting (we have measured the BoNT/A cleavage activity over two months from the inoculation) neuroprotective and anti-inflammatory actions of BoNT/A modulate the glial scar formation, reduce the astrocytes hyperactivity (as previously demonstrated and here confirmed), prevent further damage by reducing glutamate levels (excitotoxicity), and facilitate axonal regeneration. Overall, spinal tissues collected at D35 post-injury from animals treated with the toxin showed few signs of inflammation and were never severe as tissues from saline-treated mice, so much so that these mice recover motor ability more easily than saline mice, as demonstrate also by supplementary videos.

- point 4 "It is hardly to know how many animal were used"

RE: Number of animals have been better specified. In the experiments with repeated measures, the same mouse was tested on different days. For the moderate SCI group the same mice were used for testing BMS, nociceptive thresholds and SSI.

- point 5 "The discussion needs to be expanded to cover,..."

RE: Regarding pharmacodynamics and pharmacokinetics we added several sentences at the bottom of page 7 and top of page 8. Regarding the choice of Xeomin, we decided to use Xeomin because it is the commercial formulation containing the 150 kDa BoNT/A protein, without complexing protein, more similar in the composition to our lab-purified 150 KDa BoNT/A protein, already used in our previous paper to demonstrate potentiality of botulinum neurotoxin in treatment of spinal cord injury. This argument has been further stressed in the revised version.

- point 6 "Some minor issues"

RE: Regarding minor issues a, b, c, e, f, g, and h, we thank the reviewer for its detailed reporting of errors or poor explanation. All the suggested corrections have been considered and text was appropriately changed.

Regarding minor issue d, we have inserted, directly in the figure, the acronym EPI for the spinal section involved in the epicenter of the lesion and PERI for areas bordering the impact zone.

Reviewer 3 Report

The study describes the efficacy of xeomin, with laboratory-purified botulinum neurotoxin A in treating spinal cord injury in mice. The study demonstrated that both formulations can reduce excitotoxic phenomena, glial scar, inflammation, neuropathic pain, and facilitate regeneration and motor recovery in mice with spinal cord injury. However, xeomin demonstraed less efficacy than laboratory-purified botulinum neurotoxin-A, which can be improved by adjusting the dose due to differences in formulation and pharmacodynamics. The study highlights the potential of Xeomin as a possible new treatment for spinal cord injury and calls for further research to understand its mechanism of action.

In the introduction, “Today, BoNT/A and /B are licensed for 42 treatment of several autonomic nervous system and movement disorders, such as 43 dystonias, muscle spasms, spasticity, excessive sweating, overactive urinary bladder” and "Anatomical Proposal for Botulinum Neurotoxin Injection Targeting the Platysma Muscle for Treating Platysmal Band and Jawline Lifting: A Review" please add aesthetic purposes such as glabellar lines, this has been FDA approved. Citation of article published in Toxins would be helpful “Anatomical Proposal for Botulinum Neurotoxin Injection for Glabellar Frown Lines“, Botulinum toxin market in aesthetic field is taking a big portion.

Two minor errors

Sentence 53 contains an extra "of" before "BoNT/A". It should be "we injected a single dose of BoNT/A".

In sentence 62, "keeping" should be changed to "which keeps" to make the sentence grammatically correct.

In the method section please give a definition of choosing botulinum neurotoxin.

Why did you choose the Xeomin? Why not incobotulinum neurotoxin? Is it because of most purified botulinum neurotoxin so far?

In discussion,

In sentence 185, "inequivocally" should be changed to "unequivocally" to make the sentence grammatically correct.

In the text is in line 197, where "Xeomim" is misspelled. It should be "Xeomin".

I would give major revision for the article.

Author Response

Reply to Reviewer #3

We thank the reviewer for positive consideration of our paper and for helpful comments to improve its quality. All the raised concerns have been considered and appropriately replied as follow:

- General comments

"In the introduction....please add aesthetic purpose...Citation of article published in Toxins would be helpful...."

RE: Accordingly, a sentence at the beginning of page 2 (lines 44-45) and the requested citation has been added.

- "Two minor errors"

RE: We thank the reviewer for its detailed analysis of the text; all minor errors have been corrected according to the suggestions.

- "Why did you choose Xeomin? Why not Incobotulinum neurotoxin?...."

RE: We chose Xeomin because it is the commercial formulation containing the 150 kDa BoNT/A protein, without complexing protein, more similar in the composition to our lab-purified 150 KDa BoNT/A protein, already used in our previous paper to demonstrate potentiality of botulinum neurotoxin in treatment of spinal cord injury. This argument has been further stressed in the revised version. Xeomin is a commercial name for Incobotulinum neurotoxin and being Xeomin the toxin we used in this research, we referred to its commercial name rather than its scientific name.

- "In discussion..."

RE: typos errors have been corrected as suggested.

Round 2

Reviewer 3 Report

All the answers are supplied.